# ROBUST DETERMINISTIC POLICY GRADIENT FOR DISTURBANCE ATTENUATION

## ABSTRACT

Reinforcement learning (RL) has achieved remarkable success across various control and decision-making tasks. However, RL agents often show unstable and low performance when it encounter environments with unexpected external disturbances and model uncertainty. Therefore, it is crucial to develop RL agents that can sustain stable performance under such conditions. To address this issue, this paper proposes an RL algorithm called robust deterministic policy gradient (RDPG) based on adversarial RL and $H_\infty$ control methods. We formulate a maxmin objective function motivated by $H_\infty$ control, which enables both the agent and the adversary to be trained in a stable and efficient manner. In this formulation, the user seeks a robust policy to maximize the objective function, while an adversary injects disturbances to minimize it. Furthermore, for high-dimensional continuous control tasks, we introduce robust deep deterministic policy gradient (RDDPG), which combines the robustness of RDPG with the stability and learning efficiency of deep deterministic policy gradient (DDPG). Experimental evaluations in MuJoCo environments demonstrate that the proposed RDDPG outperforms baseline algorithms in terms of robustness against both external disturbances and model parameter uncertainties.

## 1 INTRODUCTION

Deep neural networks have enabled significant advances in reinforcement learning (RL) by serving as powerful function approximators, enabling RL agents to find optimal solutions in complex, high-dimensional environments such as games (Silver et al., 2017; Mnih et al., 2015) and nonlinear control systems (Lillicrap et al., 2019; Kalashnikov et al., 2018). Despite these successes, RL agents are often sensitive to external disturbances and model uncertainties (Pinto et al., 2017; Zhai et al., 2022; Vinitsky et al., 2020). In practice, physical systems inevitably face unmodeled dynamics, parameter variations, and environmental perturbations. Such factors frequently lead to unstable behavior and significant degradation in performance when deploying RL policies. Consequently, enhancing the robustness of RL algorithms against disturbances and uncertainties has emerged as a critical challenge for their reliable performance.

To address this challenge, we propose the robust deterministic policy gradient (RDPG) algorithm, which enhances the robustness of the deterministic policy gradient (DPG) framework (Silver et al., 2014) by incorporating principles from both $H_\infty$ control (Basar, 1989; Stoorvogel & Weeren, 2002) and adversarial RL (Pinto et al., 2017; Tessler et al., 2019). This novel combination is motivated by their complementary strengths and their shared foundation in a two-player zero-sum game. The $H_\infty$ control control is a classical robust control approach that provides rigorous, worst-case performance guarantees by reformulating the control problem as a min-max optimization (Başar & Bernhard, 2008; Morimoto & Doya, 2005). The controller (user) minimizes a cost function while an external disturbance (adversary) seeks to maximize it. On the other hand, adversarial RL offers a data-driven framework for building resilience against a wide range of uncertainties (Pinto et al., 2017; Tessler et al., 2019). In this context, a user agent and an adversary are trained in a zero-sum game: the user aims to maximize its long-term reward, while the adversary learns to generate the most effective noises to minimize it. This competition forces the agent to develop a robust and resilient policy.

By unifying the mathematical robustness of $H_\infty$ control with the empirical adaptability of adversarial RL under the shared structure of a two-player zero-sum game, RDPG produces control policies

that are both stable under severe disturbances and resilient to model uncertainties. Building on this unified perspective, we formalize the framework through a new objective function inspired by $H_\infty$ control (Başar & Bernhard, 2008; Morimoto & Doya, 2005), which enables the joint training of the user and the adversary. Using this objective, we develop an RL approach grounded in the DPG method Silver et al. (2014), where the user learns a policy that maximizes the objective function while the adversary simultaneously learns a policy that minimizes it, thereby establishing a two-player zero-sum learning process.

Furthermore, we extend RDPG to high-dimensional continuous control tasks by embedding it within a deep RL framework. The resulting algorithm, termed robust deep DPG (RDDPG), integrates the robustness of RDPG with the stability and learning efficiency of DDPG (Lillicrap et al., 2019). Finally, we evaluate the effectiveness of RDDPG through comprehensive experiments in MuJoCo simulation environments (Todorov et al., 2012). The results demonstrate that RDDPG not only achieves enhanced robustness against external disturbances compared to baseline algorithms, but also maintains stable performance under model parameter variations such as changes of mass and friction in systems.

## 2 PRELIMINARIES

### 2.1 ADVERSARIAL REINFORCEMENT LEARNING

Adversarial reinforcement learning can be expressed as a two-player $\gamma$ discounted zero-sum Markov game (Perolat et al., 2015). The Markov decision process (MDP) of this game can be expressed as a tuple $(\mathcal{S}, \mathcal{A}_1, \mathcal{A}_2, P, r, \gamma, s_0)$. In the MDP, $\mathcal{S}$ is the state space, $\mathcal{A}_1$ and $\mathcal{A}_2$ are the continuous action spaces for the first and second agents, respectively. The first agent selects an action $a_1 \in \mathcal{A}_1$ and the second agent selects its action $a_2 \in \mathcal{A}_2$ at the current state $s \in \mathcal{S}$ simultaneously. Then the state transits to the next state $s' \in \mathcal{S}$ with the state transition probability $P(s'|s, a_1, a_2)$ and the reward $r \in \mathbb{R}$ is incurred by the reward function $r(s, a_1, a_2, s') : \mathcal{S} \times \mathcal{A}_1 \times \mathcal{A}_2 \times \mathcal{S} \to \mathbb{R}$. For convenience, we consider a deterministic reward function and simply write $r_{k+1} := r(s_k, a_{1_k}, a_{2_k}, s_{k+1}), k \in \{0, 1, \ldots\}$. The $\gamma \in (0, 1]$ is the discounted faction and $s_0 \in \mathcal{S}$ represents the initial state. If the policy of the first and second agents are $\pi : \mathcal{S} \to \mathcal{A}_1$ and $\mu : \mathcal{S} \to \mathcal{A}_2$ respectively, the objective of the first and second agents are to maximize and minimize the cumulative discounted rewards over infinite time horizon $J^{\pi,\mu} = \mathbb{E}\left[\sum_{k=0}^{\infty} \gamma^k r_{k+1} \,\middle|\, \pi, \mu\right]$, where $\mathbb{E}[\cdot|\pi, \mu]$ is an expectation conditioned on the two policies $\pi$ and $\mu$. Perolat et al. (2015) demonstrated that for a game with optimal equilibrium return $J^*$, there always exists the Nash equilibrium, and it is equivalent to the minimax solution,

$$J^* = \min_\mu \max_\pi J^{\pi,\mu} = \max_\pi \min_\mu J^{\pi,\mu}. \tag{1}$$

### 2.2 $H_\infty$ CONTROL

As mentioned earlier, $H_\infty$ control can be formulated as a two-player zero-sum dynamic game, which enables the design of controllers that are robust against external disturbances (Başar & Bernhard, 2008). This perspective naturally extends to adversarial RL, where the agent and the adversary correspond to the two players in the game, ensuring a more stable learning process (Morimoto & Doya, 2005; Long et al., 2024). Building on this connection, we briefly introduce the fundamentals of $H_\infty$ control.

Let us consider the discrete time nonlinear discrete-time system

$$\begin{aligned} s_{k+1} &= f(s_k, a_k, w_k, v_k) \\ o_k &= g(s_k, a_k, w_k) \end{aligned} \tag{2}$$

where $s_k \in \mathbb{R}^p$ is the state, $o_k \in \mathbb{R}^q$ is the output, $a_k \in \mathbb{R}^m$ is the control input, $w_k \in \mathbb{R}^n$ is the disturbance, $v_k \in \mathbb{R}^l$ is the process noise, $f(\cdot)$ is the system dynamics function, and $g(\cdot)$ is the output function. Using the state-feedback controller $a = \pi(x)$, the system can be reduced to the autonomous closed-loop system

$$\begin{aligned} s_{k+1} &= f(s_k, \pi(x_k), w_k, v_k) \\ o_k &= g(s_k, \pi(x_k), w_k) \end{aligned}$$

Assume that the initial state $s_0$ is determined by $s_0 \sim \rho(\cdot)$, where $\rho$ is the initial state distribution. Defining the stochastic processes $\mathbf{w}_{0:\infty} := (w_0, w_1, \dots)$ and $\mathbf{o}_{0:\infty} := (o_0, o_1, \dots)$, the system can be seen as a stochastic mapping from $\mathbf{w}_{0:\infty}$ to $\mathbf{o}_{0:\infty}$ as follows: $\mathbf{o}_{0:\infty} \sim T_\pi(\cdot|\mathbf{w}_{0:\infty})$, where $T_\pi$ is the conditional probability of $\mathbf{y}_{0:\infty}$ given $\mathbf{w}_{0:\infty}$. Moreover, defining the $L^2$ norm for the general stochastic process $\mathbf{z}_{0:\infty} := (z_0, z_1, \dots)$ by

$$\|\mathbf{z}_{0:\infty}\|_{L^2} := \sqrt{\sum_{k=0}^{\infty} \mathbb{E}[\|z_k\|_2^2]},$$

The $H_\infty$ norm (Başar & Bernhard, 2008) of the autonomous system is defined as:

$$\|T_\pi\|_\infty := \sup_{\mathbf{w}_{0:\infty} \neq 0} \frac{\|\mathbf{o}_{0:\infty}\|_{L^2}}{\|\mathbf{w}_{0:\infty}\|_{L^2}} \tag{3}$$

The goal of $H_\infty$ control is to design a control policy $\pi$ that minimizes the $H_\infty$ norm of the system $T_\pi$

$$\pi^* := \arg\min_\pi \|T_\pi\|_\infty.$$

Başar & Bernhard (2008) reformulates this minimization problem as a min–max problem in a two-player zero-sum game and demonstrates that the solution of this min–max problem is equivalent to the original solution in linear time-invariant system. In this formulation, the second player (adversary) seeks to maximize the $H_\infty$ norm with its policy $\mu$, while the first player (user) aims to find a control policy $\pi$ that minimizes it

$$\pi^* = \arg\max_\pi \min_\mu \|T_\pi\|_\infty, \quad \mu^* = \arg\min_\mu \|T_\pi\|_\infty.$$

This two-player zero-sum game formulation serves as the foundation for connecting robust control theory with adversarial RL. Building on this connection, we integrate the theoretical principles of robust $H_\infty$ control into the objective function of adversarial RL and propose the robust deterministic policy gradient (RDPG) algorithm, which jointly trains a user and an adversary. In this framework, the adversary learns to generate worst-case disturbances, while the user simultaneously learns an optimal policy that remains effective under such conditions.

## 3 METHOD

In this section, we will describe the proposed method, robust deterministic policy gradient (RDPG) and its deep reinforcement learning version, robust deep deterministic policy gradient (RDDPG). We begin by formulating our problem as a two-player zero-sum Markov game between the user and the adversary. This Markov game can be regarded as a special case of the MDP introduced in Section 2.1, the user selects an action $a \in \mathcal{A}_1 = \mathbb{R}^m$ and the adversary selects disturbance $w \in \mathcal{A}_2 = \mathbb{R}^n$ at the current state $s \in \mathcal{S} = \mathbb{R}^p$ simultaneously following their the parameterized deterministic policies

$$a = \pi_\theta(s) \in \mathcal{A}_1 = \mathbb{R}^m, \quad w = \mu_\phi(s) \in \mathcal{A}_2 = \mathbb{R}^n, \quad s \in \mathcal{S} = \mathbb{R}^p$$

where $\pi_\theta := \mathbb{R}^p \to \mathbb{R}^m$ denotes the user's control policy parameterized by $\theta$, and $\mu_\phi : \mathbb{R}^p \to \mathbb{R}^n$ denotes the adversary's policy parameterized by $\phi$. In this setting, the user and adversary update their own policies to maximize and minimize the objective function, respectively, as will be described in the subsequent sections.

### 3.1 ROBUST DETERMINISTIC POLICY GRADIENT

As mentioned in Section 2.2, $H_\infty$ control enables to design controllers that are robust to external disturbances by minimizing the $H_\infty$ norm of the system $\|T_\pi\|_\infty$ like Equation 3. When incorporated into adversarial RL, $\|T_\pi\|_\infty$ is redefined as the $H_\infty$ norm of the system with respect to the user's policy $\pi$ and the adversary's policy $\mu$, denoted by $\|T_{\pi,\mu}\|_\infty$

$$\|T_{\pi,\mu}\|_\infty = \frac{\|\mathbf{o}_{0:\infty}\|_{L^2}}{\|\mathbf{w}_{0:\infty}\|_{L^2}} = \frac{\mathbb{E}\left[\sum_{k=0}^{\infty} \gamma^k r_{k+1} \mid \pi_\theta, \mu_\phi\right]}{\mathbb{E}\left[\sum_{k=0}^{\infty} \gamma^k \|w_k\|_2^2 \mid \mu_\phi\right]}.$$

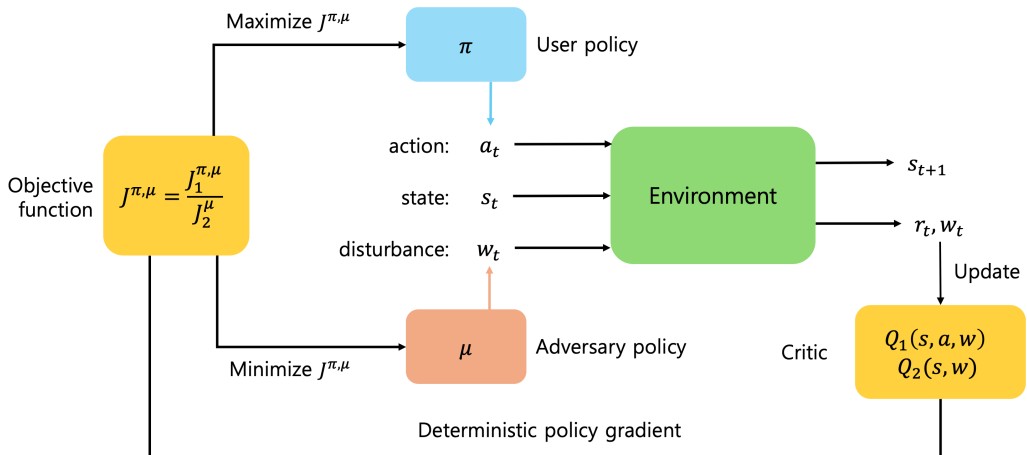

Figure 1: Overview of RDPG. Two players, the user and adversarial agents, interact in environment generating action $a_t$ and $w_t$. The action-value function $Q_1(s, a, w)$ and $Q_2(s, w)$ are updated by the reward $r_t$ and $w_t$. The policy of user $\pi$ is updated to maximize the objective function $J^{\pi,\mu}$ while the policy of adversary $\mu$ is updated to minimize it.

This leads to the definition of the objective function $J^{\pi_\theta, \mu_\phi}$, where the user's goal is to maximize the function, and the adversary's goal is to minimize it.

$$J^{\pi_\theta^*, \mu_\phi^*} = \max_\theta \min_\phi J^{\pi_\theta, \mu_\phi} = \max_\theta \min_\phi \|T_{\pi,\mu}\|_\infty = \max_\theta \min_\phi \frac{\mathbb{E}\left[\sum_{k=0}^\infty \gamma^k r_{k+1} \mid \pi_\theta, \mu_\phi\right]}{\mathbb{E}\left[\sum_{k=0}^\infty \gamma^k \|w_k\|_2^2 \mid \mu_\phi\right]}.$$

We can reformulate $J^{\pi_\theta, \mu_\phi}$ as the ratio of two terms: $J^{\pi_\theta, \mu_\phi} = J_1^{\pi_\theta, \mu_\phi} / J_2^{\mu_\phi}$ where

$$J_1^{\pi_\theta, \mu_\phi} = \mathbb{E}\left[\sum_{k=0}^\infty \gamma^k r_{k+1} \,\middle|\, \pi_\theta, \mu_\phi\right], \quad J_2^{\mu_\phi} = \mathbb{E}\left[\sum_{k=0}^\infty \gamma^k \|w_k\|_2^2 \,\middle|\, \mu_\phi\right]$$

Notice that $J_1^{\pi_\theta, \mu_\phi}$ is an objective that depends on both the user's policy $\pi_\theta$ and the adversary's policy $\mu_\phi$, whereas $J_2^{\mu_\phi}$ is a term that depends only on the adversary's policy $\mu_\phi$.

However, directly differentiating the objective function $J^{\pi_\theta, \mu_\phi} = J_1^{\pi_\theta, \mu_\phi} / J_2^{\mu_\phi}$, is challenging, making the closed-form computation of its gradient not straightforward. To simplify this computation, we employ the log-derivative trick. This formulation allows us to separate the contributions of the numerator ($J_1^{\pi_\theta, \mu_\phi}$) and denominator ($J_2^{\mu_\phi}$), thereby simplifying the computation of gradients.

$$\max_\theta \min_\phi \ln\left(\frac{J_1^{\pi_\theta, \mu_\phi}}{J_2^{\mu_\phi}}\right) = \max_\theta \min_\phi (\ln J_1^{\pi_\theta, \mu_\phi} - \ln J_2^{\mu_\phi})$$

Since the logarithm is only defined for positive arguments in this context, both $J_1^{\pi_\theta, \mu_\phi}$ and $J_2^{\mu_\phi}$ must be strictly positive to apply $\ln(\cdot)$ and the log-derivative trick. Under this condition, the gradients with respect to the user's parameters $\theta$ and the adversary's parameters $\phi$ can be expressed as follows:

$$\nabla_\theta \left\{ \ln J_1^{\pi_\theta, \mu_\phi} - \ln J_2^{\mu_\phi} \right\} = \frac{\nabla_\theta J_1^{\pi_\theta, \mu_\phi}}{J_1^{\pi_\theta, \mu_\phi}}$$

$$\nabla_\phi \left\{ \ln J_1^{\pi_\theta, \mu_\phi} - \ln J_2^{\mu_\phi} \right\} = \frac{\nabla_\phi J_1^{\pi_\theta, \mu_\phi}}{J_1^{\pi_\theta, \mu_\phi}} - \frac{\nabla_\phi J_2^{\mu_\phi}}{J_2^{\mu_\phi}}$$

Using these gradient expressions, the user updates its policy parameters via gradient ascent, while the adversary updates its parameters via gradient descent

$$\theta_{k+1} = \theta_k + \alpha_\theta \left. \frac{\nabla_\theta J_1^{\pi_\theta, \mu_{\phi_k}}}{J_1^{\pi_{\theta_k}, \mu_{\phi_k}}} \right|_{\theta = \theta_k}$$

$$\phi_{k+1} = \phi_k - \alpha_\phi \left( \frac{\nabla_\phi J_1^{\pi_{\theta_k}, \mu_\phi}}{J_1^{\pi_{\theta_k}, \mu_{\phi_k}}} - \frac{\nabla_\phi J_2^{\mu_\phi}}{J_2^{\mu_{\phi_k}}} \right) \Bigg|_{\phi = \phi_k}$$

where $\alpha_\theta$ and $\alpha_\phi$ denote the respective learning rates.

Finally, following the deterministic policy gradient theorem (Silver et al., 2014), each gradient term can be explicitly computed using the action-value function $Q_1^{\psi_1}(s, a, w)$ and $Q_2^{\psi_2}(s, w)$ . This provides a practical way to estimate gradients through sampled trajectories:

$$\nabla_\theta J_1^{\pi_\theta, \mu_\phi} = \mathbb{E}\left[ \nabla_\theta Q_1^{\psi_1}(s, \pi, \mu_\phi)\Big|_{\pi = \pi_\theta} \Bigg| s \sim \rho \right]$$

$$\nabla_\phi J_1^{\pi_\theta, \mu_\phi} = \mathbb{E}\left[ \nabla_\phi Q_1^{\psi_1}(s, \pi_\theta, \mu)\Big|_{\mu = \mu_\phi} \Bigg| s \sim \rho \right]$$

$$\nabla_\phi J_2^{\mu_\phi} = \mathbb{E}\left[ \nabla_\phi Q_2^{\psi_2}(s, \mu)\Big|_{\mu = \mu_\phi} \Bigg| s \sim \rho \right]$$

### 3.2 Robust deep deterministic policy gradient

To implement RDPG in high-dimensional continuous control tasks, we incorporate the techniques from deep deterministic policy gradient (DDPG)(Lillicrap et al., 2019), resulting in the RDDPG. In particular, we introduce the following networks:

1. The online actor network $\pi_\theta(s)$ and $\mu_\phi(s)$ for the user and the adversary respectively
2. The corresponding target networks $\pi_{\theta'}(s)$, $\mu_{\phi'}(s)$ for the two online actor network
3. Two online critic networks $Q_1^{\psi_1}(s, a, w), Q_2^{\psi_2}(s, w)$
4. The corresponding target critic networks $Q_1^{\psi_1'}(s, a, w), Q_2^{\psi_2'}(s, w)$

Now, following (Lillicrap et al., 2019), the actor and critic networks are trained through the following procedures.

#### 3.2.1 Critic update

The two critic networks $Q_1^{\psi_1}(s, a, w), Q_2^{\psi_2}(s, w)$ are trained by the gradient descent step to the loss functions $L_{\text{critic},1}(\psi_1; B)$ and $L_{\text{critic},2}(\psi_2; B)$ defined as follows:

$$L_{\text{critic},1}(\psi_1; B) := \frac{1}{|B|} \sum_{(s,a,w,r,s') \in B} (y_1 - Q_1^{\psi_1}(s, a, w))^2$$

$$L_{\text{critic},2}(\psi_2; B) := \frac{1}{|B|} \sum_{(s,a,w,r,s') \in B} (y_2 - Q_2^{\psi_2}(s, w))^2$$

where $B$ is the mini-batch, $|B|$ is the size of the mini-batch. Moreover, the target $y_1$ and $y_2$ are defined as:

$$y_1 = r + \gamma Q_1^{\psi_1'}(s', a, w)$$

$$y_2 = ||w||_2^2 + \gamma Q_1^{\psi_2'}(s', w)$$

where $r$ is the reward incurred at the same time as the state $s$, and $s'$ means the next state.

The critic's online parameters $\psi_1, \psi_2$ are updated by the gradient descent step to minimize the loss

$$\psi_i \leftarrow \psi_i - \alpha_{\text{critic}} \nabla_{\psi_i} L_{\text{critic}}(\psi_i; B), \quad i \in \{1, 2\}.$$

where $\alpha_{\text{critic}}$ is the learning rate for critic update. After the online critics are updated, the target parameters of two critic $\psi'_1$ and $\psi'_2$ are updated as follows:

$$\psi'_i \leftarrow \tau\psi_i + (1-\tau)\psi'_i, \quad i \in \{1, 2\}$$

where $\tau \in (0, 1)$ serves as the interpolation coefficient, facilitating a gradual update that enhances training stability.

### 3.2.2 Actor update

The joint actor loss equation for the user and the adversary $L_{\text{actor}}(\theta, \phi; B)$ is defined as follows:

$$L_{\text{actor}}(\theta, \phi; B) := \frac{1}{|B|} \sum_{(s,a,w,r,s') \in B} \left[ \frac{Q_1^{\psi_1}(s, \pi_\theta(s), \mu_\phi(s))}{M(Q_1^{\psi_1}) + \epsilon} - \frac{Q_2^{\psi_2}(s, \mu_\phi(s))}{M(Q_2^{\psi_2}) + \epsilon} \right]$$

where $M(Q_1^{\psi_1}) = \frac{1}{|B|}\sum_{(s,a,w,r,s') \in B} Q_1^{\psi_1}(s, a, w)$ and $M(Q_2^{\psi_2}) = \frac{1}{|B|}\sum_{(s,a,w,r,s') \in B} Q_2^{\psi_2}(s, w)$ are mean value of Q-values in batch, and $\epsilon > 0$ is a coefficient to stabilize training, and it prevents the actor's gradient from being flipped when the critic's Q-value becomes negative.

The actor networks for the user $\pi_\theta$ and adversary $\mu_\phi$ are updated using the sampled deterministic policy gradient (Silver et al., 2014; Lillicrap et al., 2019),

$$\theta \leftarrow \theta + \alpha_{\text{user}}\nabla_\theta L_{\text{actor}}(\theta, \phi; B), \quad \phi \leftarrow \phi - \alpha_{\text{adv}}\nabla_\phi L_{\text{actor}}(\theta, \phi; B)$$

where $\alpha_{\text{user}}$ and $\alpha_{\text{adv}}$ are the learning rate for the actor networks of user and adversary. Please note the opposite signs of the gradients for the user and adversary updates, which are due to their opposite roles. After the online actors are updated, the target parameters of actor $\theta'$ and $\phi'$ are updated as follows

$$\theta' \leftarrow \tau\theta + (1-\tau)\theta', \quad \phi' \leftarrow \tau\phi + (1-\tau)\phi'$$

where $\tau \in (0, 1)$ is the interpolation coefficient.

### 3.2.3 Exploration

To enable exploration, we perturb both the user and adversary policies using temporally correlated Ornstein-Uhlenbeck (OU) process (Uhlenbeck & Ornstein, 1930), following the standard DDPG approach (Lillicrap et al., 2019):

$$a_k = \pi_\theta(x_k) + \xi_k^a,$$
$$w_k = \mu_\phi(x_k) + \xi_k^w,$$

where the temporal noise terms $\xi_k^a$ and $\xi_k^w$ evolve according to the stochastic difference equation:

$$\xi_{k+1} = \xi_k + \theta_\xi(\mu_\xi - \xi_k)\Delta t + \sigma_\xi\sqrt{\Delta t}, \epsilon_k$$

where $\epsilon_k$ is a random variable drawn from a standard normal distribution $\mathcal{N}(0, 1)$. The behavior of the process is governed by the parameters $\theta_\xi$, which controls the rate of mean-reversion, $\mu_\xi$ represents the long-term mean, and $\sigma_\xi$ determines the noise scale.

The overall algorithm is described at Algorithm 1.

## 4 Experiment and results

### 4.1 Experiment setup

In this environment, we conducted two sets of experiments to evaluate the robustness of the proposed algorithm. The first experiment assesses robustness against external disturbances. Specifically, the agent is tested under random disturbances that are stronger than those used during training, in order to examine its ability to maintain performance under unseen and harsher conditions. The second experiment focuses on robustness to model parameter variations and uncertainties. In this case, the physical parameters of the model, such as mass and friction coefficients, are perturbed to simulate model mismatch, and we evaluate how well the learned policy adapts to these changes. Experiments are conducted in four MuJoCo environments, whose detailed descriptions are also presented in the Appendix A.1. We compare RDDPG with the baseline algorithms DDPG (Lillicrap et al., 2019) and RARL (Pinto et al., 2017). All algorithms are trained for 5000 episodes across ten seeds, with their hyperparameters and training settings provided in the Appendix A.2.

---

**Algorithm 1** Robust deterministic deep policy gradient (RDDPG)

---

1: Initialize the online critic networks $Q_1^{\psi_1}, Q_2^{\psi_2}$
2: Initialize the online actor networks $\pi_\theta, \mu_\phi$ for the user and adversary, respectively.
3: Initialize the target parameters $\psi_1' \leftarrow \psi_1, \psi_2' \leftarrow \psi_2, \theta' \leftarrow \theta, \phi' \leftarrow \phi$
4: Initialize the replay buffer $\mathcal{D}$
5: **for** Episode $i = 1, 2, ...N_{iter}$ **do**
6:     Observe the initial state $s_0$
7:     **for** Time step $k = 0, 1, 2, ...T - 1$ **do**
8:         Select actions $a_k = \pi_\theta(s_k) + \xi_k^a$ and $w_k = \mu_\phi(s_k) + \xi_k^w$,
9:         where $\xi_k^a, \xi_k^w$ are Ornstein-Uhlenbeck (OU) noise for exploration.
10:        Observe the next state $s_{k+1}$ and compute the reward $r_{k+1} := r(s_k, a_k, w_k, s_{k+1})$
11:        Store the transition tuple $(s_k, a_k, w_k, r_{k+1}, s_{k+1})$ in the replay buffer $\mathcal{D}$
12:        Uniformly sample a mini-batch $B$ from the replay buffer $\mathcal{D}$
13:        Update critic network:

$$\psi_i \leftarrow \psi_i - \alpha_{\text{critic}} \nabla_{\psi_i} L_{\text{critic}}(\psi_i; B), \quad i \in \{1, 2\}$$

14:        Update actor networks by the deterministic policy gradient:

$$\theta \leftarrow \theta + \alpha_{\text{user}} \nabla_\theta L_{\text{actor}}(\theta, \phi; B)$$
$$\phi \leftarrow \phi - \alpha_{\text{adv}} \nabla_\phi L_{\text{actor}}(\theta, \phi; B)$$

15:        Soft update target networks:

$$\theta' \leftarrow \tau\theta + (1 - \tau)\theta', \quad \phi' \leftarrow \tau\phi + (1 - \tau)\phi'$$
$$\psi_i' \leftarrow \tau\psi_i + (1 - \tau)\psi_i', \quad i \in \{1, 2\}$$

16:     **end for**
17: **end for**

---

## 4.2 RESULTS

### 4.2.1 ROBUSTNESS TO DISTURBANCES

| | HalfCheetah | Hopper | InvertedPendulumn | Walker2D |
|---|---|---|---|---|
| DDPG | $3160.4 \pm 2080.6$ | $1111.2 \pm 1064.5$ | $362.7 \pm 393.0$ | $841.6 \pm 886.6$ |
| RARL | $2775.8 \pm 1806.6$ | $1003.6 \pm 707.2$ | $\mathbf{795.7 \pm 369.0}$ | $916.1 \pm 978.8$ |
| RDDPG | $\mathbf{3306.2 \pm 1417.0}$ | $\mathbf{1150.9 \pm 951.3}$ | $677.84 \pm 440.44$ | $\mathbf{1020.4 \pm 715.0}$ |

Table 1: Mean and standard deviation of episode rewards across ten seeds under random disturbances. Bold indicates the highest average reward. RDDPG exhibits more robust and stable performance compared to the baselines, indicating that incorporating $H_\infty$ control into adversarial reinforcement learning enhances the robustness of deterministic policy learning against disturbances.

As shown in Table 1, the proposed RDDPG algorithm achieves a higher mean episode reward and a lower standard deviation compared to DDPG algorithms under random disturbances. This result suggests that RDDPG enhances robustness to external disturbances by leveraging an adversarial RL framework that formulates the problem as a two-player zero-sum game between the agent and the adversary. Through this dynamic interaction, the agent is compelled to learn a more generalized and robust policy rather than overfitting to specific training conditions. The lower variance of RDDPG compared to DDPG further highlights that its learned policy is more consistent and stable, avoiding significant degradation under unmodeled or unexpected disturbances.

RARL also demonstrates improved robustness against disturbances compared to DDPG. However, as shown in Table 1, our proposed RDDPG algorithm achieves a higher performance except InvertedPendulumn. A key distinction lies in the objective function While RARL focuses on training the user to maximize the cumulative reward $\mathbb{E}\left[\sum_{k=0}^{\infty} \gamma^k r_{k+1}\right]$ and the adversary to simply minimize it, RDDPG integrates $H_\infty$ control theory into the adversarial RL framework to effectively train both

the user and the adversary. By embedding the $H_\infty$ objective function into our objective function $J^{\pi_\theta, \mu_\phi} = J_1^{\pi_\theta, \mu_\phi} / J_2^{\mu_\phi}$, RDDPG guides the user agent to learn a policy that is optimally robust against the most challenging disturbances. Concurrently, the adversary agent is trained to generate precisely this worst-case disturbance that most effectively hinders the user's policy.

### 4.2.2 ROBUSTNESS TO MODEL UNCERTAINTY

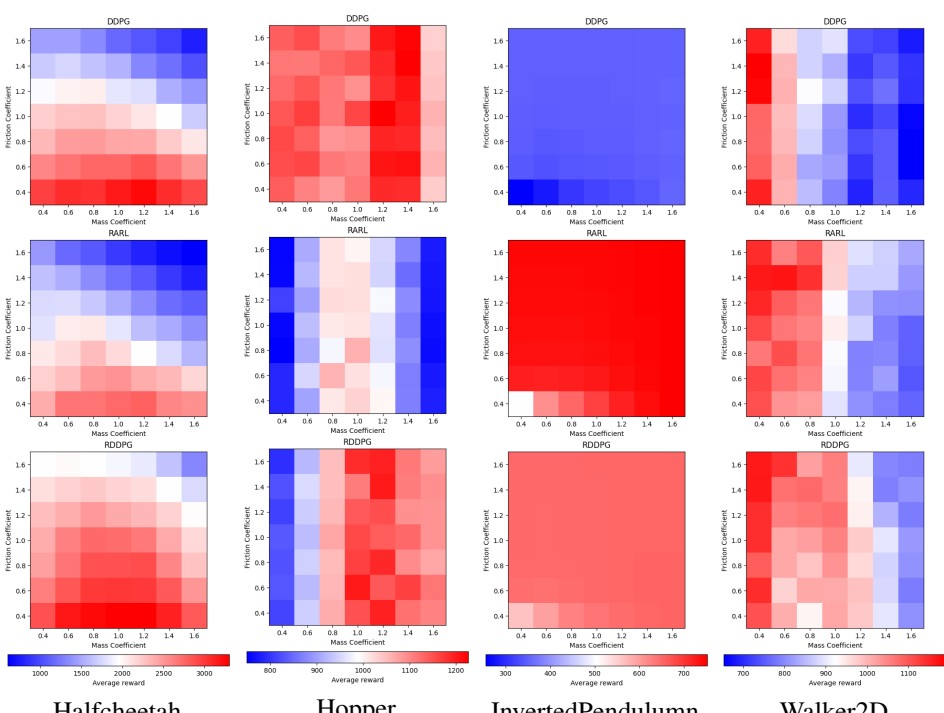

Figure 2: Heatmap of mean episode rewards across ten seeds under random disturbances. Red indicates higher rewards, while blue indicates lower rewards. The rows correspond to the algorithms, with the first, second, and third rows representing DDPG, RARL, and RDDPG, respectively. The columns correspond to the environments, which are HalfCheetah, Hopper, InvertedPendulum, and Walker2D. In each heatmap, the horizontal axis represents the mass coefficient, while the vertical axis represents the friction coefficient.

Figure 2 presents a visualization of each algorithm's performance, evaluated across a grid of mass and friction coefficient variations for four environments. These heatmaps vividly illustrate differences in their robustness to model uncertainty and disturbances. Across all plots, red signifies higher average rewards, while blue indicates lower performance.

Except Hopper environment, RDDPG maintains stable performance across different mass and friction conditions, while DDPG displays significant fluctuations with large areas of low rewards, indicating lower robustness to parameter changes. Compared to RARL, RDDPG achieves consistently higher rewards under model parameter variations. This implies that the RDPG formulation enables the adversary to learn more effective and challenging disturbance strategies, which in turn drives the user to develop more robust policies against model uncertainty.

## 5 RELATED WORKS

Robust reinforcement learning (RRL) is proposed to address the limitations of standard RL by explicitly accounting for model inaccuracies and external disturbances (Morimoto & Doya, 2005). By formulating the problem as a differential game within the $H_\infty$ control framework, the actor learns optimal control inputs while anticipating the worst-case disturbance. This approach demonstrates

superior robustness on a nonlinear inverted pendulum task, where it successfully handles environmental changes that cause standard RL controllers to fail.

Building on these ideas, recent work extends RRL into the deep reinforcement learning (DRL) setting to improve robustness under various scenarios (Pinto et al., 2017; Zhai et al., 2022; Long et al., 2024; Tessler et al., 2019; Lee & Lee, 2025; Vinitsky et al., 2020; Dong et al., 2023). Robust adversarial reinforcement learning (RARL), inspired by RRL and $H_\infty$ control, trains a protagonist agent to operate effectively while an adversary agent learns to apply the worst-case disturbances (Pinto et al., 2017). This two-player zero-sum game formulation yields policies that are highly robust to varying test conditions. However, training both agents in a balanced manner is challenging. Since the adversary's policy often learns faster than the protagonist's, it is easier to develop a strong adversary than to obtain a stable control policy. An excessively dominant adversary can destabilize the system, bias the sampling process, undermine learning stability, and even degrade the robustness of the resulting policy.

To address this issue, Zhai et al. (2022) propose a dissipation-inequality-constrained adversarial RL framework that ensures system stability during training. By extending the dissipative principle of robust $H_\infty$ control to Markov Decision Processes, they derive stability constraints based on $L_2$-gain performance. Similarly, Long et al. (2024) presents a novel approach to enhance a robot's ability to withstand external disturbances. The authors frame the policy learning process as an adversarial interaction between the robot's locomotion policy and a learnable disturber that generates the most destabilizing forces and their method incorporates an $H_\infty$ constraint to maintain the stability of the joint optimization. Previous approaches often require additional computation for updating constraints, which can significantly increase training complexity. In contrast, our work introduces a simple objective function, derived from the $H_\infty$ control framework, that eliminates the need for constraint updates and enables training to proceed using only the interaction between the user and the adversary, thereby simplifying the learning process. Furthermore, these prior methods concentrate on improving the robustness of on-policy stochastic policies, such as TRPO (Schulman et al., 2015) and PPO (Schulman et al., 2017). In this paper, we demonstrate that RDPG extends robustness improvements to off-policy deterministic policies. This not only enhances robustness against disturbances and model uncertainties but also achieves stability and sample efficiency compared to on-policy algorithms in continuous action spaces.

Lee & Lee (2025) formulates $H_\infty$ control problem into two-player zero-sum dynamic game and leverages cost function to update both user and adversary. However, the performance varies depending on a coefficient in the cost function, which controls the influence of disturbances, and finding an optimal value for the coefficient is challenging. In contrast, RDPG allows for the training of both agents without the need to tune the coefficient.

Tessler et al. (2019) propose the action-robust Markov decision process (AR-MDP), where the adversary directly interferes with the agent's actions by either replacing them or adding perturbations. In contrast, our approach operates in a different manner: the adversary does not modify the user's action. Instead, the adversary injects disturbances directly into the system dynamics, while both the user and the adversary are jointly trained to maximize and minimize the objective function.

Vinitsky et al. (2020); Dong et al. (2023) argue that a single adversary can get stuck in a local optimum, which can lead to poor robustness performance. Therefore, they employ a group of adversaries in adversarial RL and optimize the average performance to improve the robustness. In contrast, our work incorporate well-established robust control theory, $H_\infty$, into the adversarial RL to prevent degraded robustness performance.

## 6 CONCLUSION

This paper proposes RDPG, which combines the concept of the $H_\infty$ control problem and adversarial RL, to overcome the robustness problem of DRL algorithms. Furthermore, we propose the RDDPG, which combines the robustness framework of RDPG with the stability and efficiency of DDPG, thereby achieving enhanced disturbance attenuation and stable policy learning in continuous control tasks. We evaluate RDDPG in several Mujoco environment and the results show that RDDPG successfully learns an optimal control policy and outperforms other DRL algorithms while maintaining stable performance across different disturbance and model parameters scenarios.

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
