## A    EXPERIMENTS DETAILS

### A.1    ENVIRONMENTS

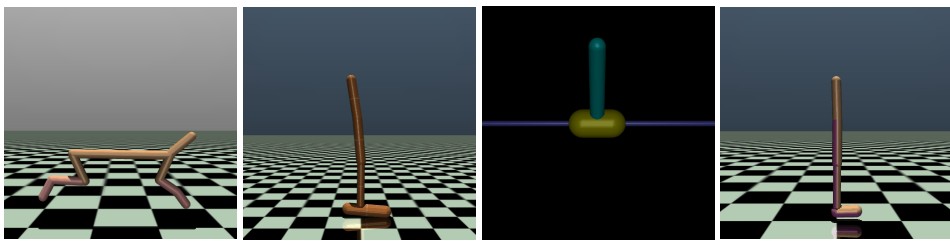

Figure 3: Experiment environments. From left to right: (a) HalfCheetah, (b) Hopper, (c) Inverted-Pendulum, and (d) Walker2D

(a) HalfCheetah

The HalfCheetah is a two-dimensional simulated robot consisting of 9 body parts and 8 joints. The environment provides a 17-dimensional state space, which includes joint positions and velocities, and a 6-dimensional continuous action space, corresponding to the torques applied to the controllable joints.

(b) Hopper

The Hopper is a two-dimensional robot consisting of 4 body parts and 3 controllable joints. The environment provides an 11-dimensional state space, which includes joint positions and velocities, and a 3-dimensional continuous action space corresponding to the torques applied at the thigh, leg, and foot joints.

(c) InvertedPendulum

The InvertedPendulum consists of a cart moving along a one-dimensional track with a pendulum attached to it. The environment provides a 4-dimensional state space, including the cart position/velocity and pendulum angle/angular velocity, and a 1-dimensional continuous action space representing the force applied to the cart.

(d) Walker2D

The Walker2d is a two-dimensional biped robot composed of 7 body parts and 6 controllable joints. The environment provides a 17-dimensional state space, which includes joint positions and velocities, and a 6-dimensional continuous action space corresponding to the torques applied at the joints of the legs.

|  | (a) | (b) | (c) | (d) |
|---|---|---|---|---|
| The dimension of state space | 17 | 11 | 4 | 17 |
| The dimension of action space | 6 | 3 | 1 | 6 |
| The range of action space | $[-1, 1]$ | $[-1, 1]$ | $[-3, 3]$ | $[-1, 1]$ |
| The dimension of disturbance space | 3 | 2 | 1 | 3 |
| The range of disturbance space | $[-5, 5]$ | $[-5, 5]$ | $[-3, 3]$ | $[-5, 5]$ |
| The name of perturbed body | torso hfoot foot | torso foot | pole | torso lfoot foot |

Table 2: Details about state, action, disturbance spaces of each environment: (a):HalfCheetah, (b):Hopper, (c):InvertedPendulum, (d):Walker2D.

## A.2 ALGORITHM DETAILS

The hyperparameters of algorithms used in experiments are described in Table 3. The overall descriptions of DDPG and RARL are desribed in Algorithm 2 and 3

| | DDPG | RARL | RDDPG |
|---|---|---|---|
| Number of Episodes for training | 5000 | 5000 | 5000 |
| Buffer size | 1000000 | 1000000 | 1000000 |
| Learning rate for critic $\alpha_{\text{critic}}$ | 0.001 | 0.001 | 0.001 |
| Learning rate for actor of user $\alpha_{\text{user}}$ | 0.0001 | 0.0001 | 0.0001 |
| Learning rate for actor of adversary $\alpha_{\text{adv}}$ | - | 0.0001 | 0.0001 |
| Hidden layer sizes | $[256, 256]$ | $[256, 256]$ | $[256, 256]$ |
| $\tau$ | 0.005 | 0.005 | 0.005 |
| Batch size $|B|$ | 128 | 128 | 128 |
| Activation function in actor | tanh | tanh | tanh |
| Optimizer | Adam | Adam | Adam |
| Discounted factor $\gamma$ | 0.99 | 0.99 | 0.99 |
| Policy noise | OU noise | OU noise | OU noise |
| Mean of OU noise | 0 | 0 | 0 |
| Standard deviation of OU noise | 0.2 | 0.2 | 0.2 |

Table 3: Details about the hyperparameters of each algorithms

## A.3 EXPERIMENT

### A.3.1 ROBUSTNESS TO DISTURBANCES

To evaluate the robustness against external disturbances, we applied stronger perturbations than those used during training (see Table 2). Specifically, the InvertedPendulumn environment was tested under random disturbances sampled from $[-3, 3]$, while all other environments were subjected to disturbances in the range of $[-10, 10]$. For each setting, we conducted 500 test episodes across ten seeds and reported the mean and standard deviation of the cumulative reward per episode.

### A.3.2 ROBUSTNESS TO MODEL UNCERTAINTY

To assess robustness against model parameter uncertainties, we varied both the torso mass and the ground friction coefficient in the environment. Specifically, each parameter was scaled by one of the factors $[0.4, 0.6, 0.8, 1.0, 1.2, 1.4, 1.6]$, resulting in a total of 49 different scenarios. For each scenario, we evaluated the trained policies over 100 test episodes using ten seeds, and the average cumulative rewards were visualized in the form of a heatmap shown in Figure 2. Furthermore, in order to examine whether the policies remain robust under simultaneous parameter variations and external perturbations, we applied the same disturbance settings as in the previous experiment while varying the model parameters.

---

**Algorithm 2** Deterministic deep policy gradient (DDPG)

---

1: Initialize the online critic networks $Q^\psi$
2: Initialize the actor networks $\pi_\theta$ for the user.
3: Initialize the target parameters $\psi' \leftarrow \psi, \theta' \leftarrow \theta$
4: Initialize the replay buffer $\mathcal{D}$
5: **for** Episode $i = 1, 2, ...N_{iter}$ **do**
6:      Observe the initial state $s_0$
7:      **for** Time step $k = 0, 1, 2, ...T - 1$ **do**
8:          Select actions $a_k = \pi_\theta(s_k) + \xi_k^a$ and $w_k = \mu_\phi(s_k) + \xi_k^w$,
9:          where $\xi_k^a$, $\xi_k^w$ are Ornstein-Uhlenbeck (OU) noise for exploration.
10:         Observe the next state $s_{k+1}$ and compute the reward $r_{k+1} := r(s_k, a_k, s_{k+1})$
11:         Store the transition tuple $(s_k, a_k, r_{k+1}, s_{k+1})$ in the replay buffer $\mathcal{D}$
12:         Uniformly sample a mini-batch $B$ from the replay buffer $\mathcal{D}$
13:         Update critic network:

$$\psi \leftarrow \psi - \alpha_{\text{critic}} \nabla_\psi L_{\text{critic}}(\psi; B)$$

          where

$$L_{\text{critic}}(\psi; B) := \frac{1}{|B|} \sum_{(s,a,w,r,s') \in B} (r + \gamma Q^{\psi'}(s', a) - Q^\psi(s,a))^2$$

14:         Update actor networks by the deterministic policy gradient:

$$\theta \leftarrow \theta + \alpha_{\text{user}} \nabla_\theta L_{\text{actor}}(\theta; B)$$

          where

$$L_{\text{acotr}}(\theta; B) := \frac{1}{|B|} \sum_{(s,a,w,r,s') \in B} \left[ Q^\psi(s, \pi_\theta(s)) \right]$$

15:         Soft update target networks:

$$\psi' \leftarrow \tau\psi + (1 - \tau)\psi', \quad \theta' \leftarrow \tau\theta + (1 - \tau)\theta'$$

16:      **end for**
17: **end for**

---

---

**Algorithm 3** Robust adversarial reinforcement learning (RARL)

---

1: Initialize the online critic networks $Q_1^{\psi_1}, Q_2^{\psi_2}$
2: Initialize the actor networks $\pi_\theta, \mu_\phi$ for the user and adversary, respectively.
3: Initialize the target parameters $\psi_1' \leftarrow \psi_1, \psi_2' \leftarrow \psi_2, \theta' \leftarrow \theta, \phi' \leftarrow \phi$
4: Initialize the replay buffer $\mathcal{D}$
5: **for** Episode $i = 1, 2, ... N_{iter}$ **do**
6:      Observe the initial state $s_0$
7:      **for** Time step $k = 0, 1, 2, ... T - 1$ **do**
8:          Select actions $a_k = \pi_\theta(s_k) + \xi_k^a$ and $w_k = \mu_\phi(s_k) + \xi_k^w$,
9:          where $\xi_k^a$, $\xi_k^w$ are Ornstein-Uhlenbeck (OU) noise for exploration.
10:         Observe the next state $s_{k+1}$ and compute the reward $r_{k+1} := r(s_k, a_k, w_k, s_{k+1})$
11:         Store the transition tuple $(s_k, a_k, w_k, r_{k+1}, s_{k+1})$ in the replay buffer $\mathcal{D}$
12:         Uniformly sample a mini-batch $B$ from the replay buffer $\mathcal{D}$
13:         Update critic network:

$$\psi_i \leftarrow \psi_i - \alpha_{\text{critic}} \nabla_{\psi_i} L_{\text{critic}}(\psi_i; B), \quad i \in \{1, 2\}$$

where

$$L_{\text{critic},1}(\psi_1; B) := \frac{1}{|B|} \sum_{(s,a,w,r,s') \in B} (r + \gamma Q_1^{\psi_1'}(s', a) - Q_1^{\psi_1}(s, a))^2$$

$$L_{\text{critic},2}(\psi_2; B) := \frac{1}{|B|} \sum_{(s,a,w,r,s') \in B} (-r + \gamma Q_2^{\psi_2'}(s', w) - Q_2^{\psi_2}(s, w))^2$$

14:         Update actor networks by the deterministic policy gradient:

$$\theta \leftarrow \theta + \alpha_{\text{user}} \nabla_\theta L_{\text{user}}(\theta, \phi; B)$$
$$\phi \leftarrow \phi + \alpha_{\text{adv}} \nabla_\phi L_{\text{adv}}(\theta, \phi; B)$$

where

$$L_{\text{user}}(\theta; B) := \frac{1}{|B|} \sum_{(s,a,w,r,s') \in B} \left[ Q_1^{\psi_1}(s, \pi_\theta(s)) \right]$$

$$L_{\text{adv}}(\phi; B) := \frac{1}{|B|} \sum_{(s,a,w,r,s') \in B} \left[ Q_2^{\psi_2}(s, \mu_\phi(s)) \right]$$

15:         Soft update target networks:

$$\theta' \leftarrow \tau\theta + (1 - \tau)\theta', \quad \phi' \leftarrow \tau\phi + (1 - \tau)\phi'$$
$$\psi_i' \leftarrow \tau\psi_i + (1 - \tau)\psi_i', \quad i \in \{1, 2\}$$

16:      **end for**
17: **end for**

---