# OpenReview forum: "Robust deterministic policy gradient for disturbance attenuation"
_ICLR.cc/2026/Conference — ICLR 2026 Conference Withdrawn Submission_

### Official Review · Reviewer_9Dr9 · 2025-10-30

**Soundness:** 2
**Presentation:** 3
**Contribution:** 1
**Rating:** 2
**Confidence:** 3

**Summary:**

This paper proposes Robust Deterministic Policy Gradient (RDPG), an RL algorithm that integrates principles from control theory and adversarial RL to improve robustness against external disturbances and model uncertainty. The approach formulates the control problem as a two-player zero-sum game between a user agent (policy) and an adversary (disturbance generator), resulting in Robust Deep Deterministic Policy Gradient (RDDPG), which combines the RDPG framework with the stability and sample efficiency of DDPG.

Experiments on four MuJoCo environments show that RDDPG improves robustness to both random external disturbances and model parameter variations.

**Strengths:**

+ The work targets a practically important problem: robustness to disturbances and uncertainties in RL control systems, and connects classical robust control with modern adversarial RL.

+ The RDDPG variant is well integrated with DDPG’s structure, making it readily applicable to continuous control tasks.

+ The paper is clearly organized, with thorough background sections connecting robust control theory and RL.

**Weaknesses:**

-- Using random disturbances as the robustness test might not sufficiently challenge the agents. A more meaningful evaluation would be to identify the worst-case adversary for each method and test policies against those learned adversaries.

-- The only robust RL comparison is RARL (2017). Without more recent baselines, it is difficult to assess how competitive RDDPG truly is.

-- Evaluation is confined to four MuJoCo tasks; broader coverage (e.g., Ant or other benchmarks) would strengthen generality.

-- (minor) The legends and axis labels in Figure 2 are small and difficult to read; improving their clarity would help interpret the results.

-- Beyond the integration of control and adversarial training, the paper’s distinct algorithmic or theoretical contributions are limited. It is not clear what new insights are offered beyond the straightforward combination of two existing frameworks. The authors should emphasize why prior robust RL or adversarial methods cannot operate effectively in off-policy deterministic settings, and how RDPG specifically overcomes these limitations.

**Questions:**

Conceptually, is RARL a special case of RDPG where $J_2 = 1$? What additional benefits does the H∞ normalization introduce?

Can RDPG handle stochastic or observation-based disturbances, such as in POMDP or sensor noise settings?

---

### Official Review · Reviewer_dw81 · 2025-11-02

**Soundness:** 2
**Presentation:** 1
**Contribution:** 2
**Rating:** 2
**Confidence:** 1

**Summary:**

Paper is trying to exploit $H_\infty$ techniques for robust RL.


Honestly, I am having trouble understanding the core message of the paper, despite my reasonable experience in the Robust RL.

**Strengths:**

Using $H_\infty$ techniques for robust RL seems novel to me.  However, I don't understand the clear advantages over the traditional RL approaches [1,2,3].



[1]@inproceedings{
kumar2024efficient,
title={Efficient Value Iteration for s-rectangular Robust Markov Decision Processes},
author={Navdeep Kumar and Kaixin Wang and Kfir Yehuda Levy and Shie Mannor},
booktitle={Forty-first International Conference on Machine Learning},
year={2024},
url={https://openreview.net/forum?id=J4LTDgwAZq}
}

[2]@inproceedings{
kumar2025nonrectangular,
title={Non-rectangular Robust {MDP}s with Normed  Uncertainty Sets},
author={Navdeep Kumar and Adarsh Gupta and Maxence Mohamed ELFATIHI and Giorgia Ramponi and Kfir Yehuda Levy and Shie Mannor},
booktitle={The Thirty-ninth Annual Conference on Neural Information Processing Systems},
year={2025},
url={https://openreview.net/forum?id=Xx0cJGXU7n}
}

[3]@unknown{unknown,
author = {Zouitine, Adil and Bertoin, David and Clavier, Pierre and Geist, Matthieu and Rachelson, Emmanuel},
year = {2024},
month = {06},
pages = {},
title = {RRLS : Robust Reinforcement Learning Suite},
doi = {10.48550/arXiv.2406.08406}
}

**Weaknesses:**

Particularly, I don't understand fully the relation between $H_\infty$ control and Robust RL. Maybe authors can make little more effort to make it more clear.

Paper also misses substantial amount of relevant literature in robust RL.

**Questions:**

Q1) Can this approach be used for simple finite state-action space in the tabular setting? If so can you please, make an explicit connection between $H_\infty$ control and robust RL in this simplest setting.

Q2) The approach in the paper works for which kind of uncertainty sets among sa-rectangular, s-rectangular and non-rectangular? Does the approach takes into the account for the amount of uncertainty in the model, that is, how does the algorithm adapt for different levels of uncertainty sets?

---

### Official Review · Reviewer_njZV · 2025-11-03

**Soundness:** 1
**Presentation:** 3
**Contribution:** 1
**Rating:** 2
**Confidence:** 4

**Summary:**

This paper proposes a deep RL algorithm that is robust to environmental uncertainty. The algorithm is motivated by $H_\infty$ control theory. Empirical evaluation demonstrates the proposed method's effectiveness in several control tasks.

**Strengths:**

The idea of deriving a robust objective from $H_\infty$ control theory is interesting. The paper is well organized.

**Weaknesses:**

The paper lacks mathematical rigor in several places, which makes me question the correctness of this paper. Below I list specific issues.

---

In Section 3.1, the authors derive the $H\_\\infty$ norm of the system $\\|T\_\\pi\\|\_{\\infty}$:

$$
\\left\\|T\_{\\pi, \\mu}\\right\\|\_{\\infty}\\underbrace{=}\_{(a)}\\frac{\\left\\|\\mathbf{o}\_{0: \\infty}\\right\\|\_{L^2}}{\\left\\|\\mathbf{w}\_{0: \\infty}\\right\\|\_{L^2}}\\underbrace{=}\_{(b)}\\frac{\\mathbb{E}\\left[\\sum\_{k=0}^{\\infty} \\gamma^k r\_{k+1} \\mid \\pi\_\\theta, \\mu\_\\phi\\right]}{\\mathbb{E}\\left[\\sum\_{k=0}^{\\infty} \\gamma^k\\left\\|w\_k\\right\\|\_2^2 \\mid \\mu\_\\phi\\right]}
$$

I am confused about this derivation:

1. How is $\\mathbf{o}\_{0:\\infty}$ connected to the reward sequence $r\_{1:\\infty}$? The relationship is not defined in the text. Also, the right-hand side appears to be missing a square root.
2. After (a) the expression should include $\\sup\_w$.

---

I believe Eq. (1) is incorrect in general. To make it hold, one needs structural assumptions on the reward function $r$. For example, when $\\gamma=0$ Eq. (1) reduces to

$$
J^{*} = \\min\_\\mu \\max\_\\pi r(\\pi, \\mu) = \\max\_\\pi \\min\_\\mu r(\\pi, \\mu)\\;,
$$

which obviously does not hold for a general $r$. It does hold under special conditions, such as when $r(a\_1,a\_2)$ is concave in $a\_1$ and convex in $a\_2$, by Sion's minimax theorem.

---

About the equation in line 189:
- Is this a definition? If so, denote it with $\\triangleq$ or $\\coloneqq$ instead of $=$.
- If it is a claimed equality, why does the second equality hold?

---

Define

$$
f(\\theta, \\phi) = \\frac{\\mathbb{E}\\left[\\sum\_{k=0}^{\\infty} \\gamma^k r\_{k+1} \\mid \\pi\_\\theta, \\mu\_\\phi\\right]}{\\mathbb{E}\\left[\\sum\_{k=0}^{\\infty} \\gamma^k\\left\\|w\_k\\right\\|\_2^2 \\mid \\mu\_\\phi\\right]}.
$$

- According to line 189, the algorithm performs gradient ascent-descent on $f(\\theta,\\phi)$ to solve $\\max\_\\theta \\min\_\\phi f(\\theta,\\phi)$.
- This ascent-descent makes sense if the max-min and min-max are equivalent. However, it is non-trivial to guarantee this duality even under some structural assumptions on $r$, because the denominator depends on $\\phi$, which complicates convexity/concavity arguments.

Overall, the mathematical portion of the paper needs rigorous and clear arguments. At minimum, the authors should address the above issues.

---

Additionally, I have concerns about the novelty. It is unclear what advantages this method offers over existing robust deep RL methods, such as the work at https://arxiv.org/pdf/2205.07344. This also proposes a robust deep RL algorithm, and it has several theoretical guarantees at least in the tabular setting (see https://arxiv.org/abs/2212.10439).

**Questions:**

See weaknesses

---

### Official Review · Reviewer_CW1A · 2025-11-11

**Soundness:** 2
**Presentation:** 2
**Contribution:** 2
**Rating:** 2
**Confidence:** 3

**Summary:**

This paper studies an adversarial variant of the reinforcement learning problem, where in addition to learner adaptively choosing their action, an adversary adaptively chooses a disturbance (as a function of the state). Loosely inspired by H-infty control, the paper proposes a minimax objective that trains a policy against an adversarial agent to maximize the expected reward per unit "energy" (or l2 norm) of disturbance injected into the system. The paper then proposes a further deterministic policy (actor-critic style) algorithm that evaluates it on 4 Mujoco environments.

**Strengths:**

In general, in any minimax setting, it is important to constrain the adversary, e.g., via regularization, step size etc. The paper puts forward a reasonable alternative proposal that limits the l2 norm of the disturbances the adversary can produce.

The proposal shows reasonable empirical gains for the settings tested in the paper over RARL.

**Weaknesses:**

Despite the presentation, the connection with H-infty control is at most skin deep. Classical robust control results guarantee minimax performance against arbitrary adversary (across all policies, that is, rather than one restricted to a given policy class), and this has provable links to model uncertainty via the small gain theorem. Both these characteristics are absent here. Further, I'm not sure why line 160 follows. The numerator here looks like a l1 norm, and the denominator is a l2-squared norm, very far from the ratio of two l2 norms.

Instead, the results here are better understood in terms of how to best constrain an adversary: the paper suggests the as the sum of squares of the produced disturbances. With this framing, I would expect a more complete evaluation than compares this to parameter norm bounds, step size limits, and/or, for example, entropy regularization. The paper compares solely to Pinto et al 2017 (published 8 years ago), which has ~1K citations, and hence quite a bit of follow up work. Can the authors comment on how the paper compare to more recent adversarial RL works? And if they would be suitable for comparisons?

Starting with line 190, the work denotes the denominator purely as a function of the adversaries policies (J_2^{\mu_\phi}). But this seems like either a mistake or approximation, because the disturbances indeed are produced by \mu_\phi acting on states, but the distribution over states is jointly arrived at using both the learner's and the adversary's policies. This would dictate how one differentiates through the dynamics.

While robustness to disturbances results are mismatched in the sense they consider zero-mean disturbances, which is in the stochastic control territory rather than robust control. It would be more encouraging to see performances against "difficult" disturbances.

On a similar note, model uncertainties are well considered. But here there is a vast literature on robust MDPs (both with and without function approximation) that is tailored to these settings. But no comparisons are present.

**Questions:**

I would like the authors to address the points raised in the last section and point out any misunderstanding where necessary.

Minor comment: On line 111, y_{0:\infty} should be o_{0:\infty}.

---

### Note · Authors · 2025-11-28

I have read and agree with the venue's withdrawal policy on behalf of myself and my co-authors.